# Tracing the origins of *Plasmodium vivax* resurgence after malaria elimination on Aneityum Island in Vanuatu
Sho Sekine [1,2], Chim W. Chan[1], Morris Kalkoa[3], Sam Yamar[3], Harry Iata[3], George Taleo[3], Achyut KC [1,4], Wataru Kagaya [1,5], Yasutoshi Kido[1] & Akira Kaneko [1,4] ✉

## Abstract

**Background** Five years after successful malaria elimination, Aneityum Island in Vanuatu experienced an outbreak of *Plasmodium vivax* of unknown origin in 2002. Epidemiological investigations revealed several potential sources of *P. vivax*. We aimed to identify the genetic origin of *P. vivax* responsible for the resurgence.

**Methods** Five *P. vivax* microsatellite markers were genotyped using DNA extracted from archived blood samples. A total of 69 samples from four *P. vivax* populations was included: 29 from the outbreak in 2002, seven from Aneityum in 1999 and 2000, 18 from visitors to Aneityum in 2000, and 15 from nearby Tanna Island in 2002. A neighbour-joining phylogenetic tree was constructed to elucidate the relationships among *P. vivax* isolates. STRUCTURE and principal component analysis were used to assess patterns of genetic structure.

**Results** Here we show distinct genetic origins of *P. vivax* during the outbreak on Aneityum. While the origin of most *P. vivax* lineages found during the outbreak remains unidentified, limited genetic diversity among these lineages is consistent with a rapid expansion from a recent common ancestor. Contemporaneous *P. vivax* from neighboring Tanna and potential relapse of *P. vivax* acquired from other islands in 1999 and 2000 are also identified as minor contributors to the outbreak.

**Conclusions** Multiple reintroductions of *P. vivax* after elimination highlight the high receptivity and vulnerability to malaria resurgence in island settings of Vanuatu, despite robust surveillance and high community compliance to control measures.

## Plain language summary

*Plasmodium vivax* is one of several parasite species that cause malaria. On Aneityum Island in Vanuatu, malaria had been eliminated in 1997, but an outbreak was reported in 2002 despite protective measures still being in place. Here, we analysed DNA of parasites from the outbreak to understand its origin, since parasites of different origins will have slight differences in their DNA. Most parasites had similar DNA suggesting they had a recent shared common ancestor whose origin remains unidentified. From this analysis we were also able to find a minority of parasites that likely came from Tanna in 2002, while another small group of parasites may have originated from parasites imported to Aneityum in 1999 or 2000. This illustrates the difficulty of maintaining a malaria-free status in resource-limited areas and the threat of imported malaria to elimination efforts.

*Plasmodium vivax* is the predominant cause of malaria outside sub-Saharan Africa[1]. In countries with co-endemic *P. falciparum* and *P. vivax*, the proportion of *P. vivax* infections has consistently increased as the total malaria burden decreases, highlighting the considerable adaptive potential and relative resilience of this[2,3]. *P. vivax* infections are characterised by the dormant and persistent hypnozoite form in the liver, and its activation can cause recurrent blood-stage infections known as relapses, hampering efforts to control and eventually eliminate the species[4]. The re-emergence of *P. vivax* in many regions following successful elimination programmes serves as a crucial reminder of the propensity of the parasite to resurge in areas where it has almost been eliminated[5,6]. After malaria elimination, preventing the re-establishment of *Plasmodium* parasites and their transmission is crucial for sustaining malaria remission. As most of the eliminated areas are positioned along the margins of areas of endemicity, investigating the origins of imported parasites can help design promising targeted interventions[7].

Islands provide natural ecological experiments with considerable potential for malaria intervention studies[8] and some have demonstrated early success towards malaria elimination[9,10]. Vanuatu is an archipelago of 82 islands spanning 1176 km in the South Pacific. It straddles the Buxton

[1]Department of Virology, Graduate School of Medicine, and Osaka International Research Center for Infectious Diseases, Osaka Metropolitan University, Osaka, Japan. [2]Department of Medical Technology, Morinomiya University of Medical Sciences, Osaka, Japan. [3]National Malaria and other Vector Borne Diseases Control Program (NVBDCP), Ministry of Health, Port Vila, Vanuatu. [4]Department of Microbiology, Tumor and Cell Biology, Karolinska Institutet, Stockholm, Sweden. [5]Department of Eco-epidemiology, Institute of Tropical Medicine, Nagasaki University, Nagasaki, Japan. ✉e-mail: akira.kaneko@ki.se

line (170°E, 20°S; Fig. 1), which delineates the extension of *Anopheles* mosquitoes[11]. Malaria is endemic on most islands of Vanuatu. Both *P. falciparum* and *P. vivax* historically contributed to the overall malaria burden. Malaria incidence typically peaks in March and April, with a more conspicuous increase in *P. falciparum* than in *P. vivax*[12]. High population coverage of long-lasting insecticidal nets and widespread deployment of malaria rapid diagnostic tests and artemisinin-based combination therapies have substantially reduced malaria incidence since 2010[13]. In 2021, Vanuatu reported 312 indigenous *P. vivax* cases, corresponding to an annual parasite incidence of 0.98 cases per 1000 population[1].

Aneityum Island in Tafea Province is the southernmost inhabited island in Vanuatu and the only malaria-endemic island outside the Buxton line (Fig. 1). Tanna, which lies northwest of Aneityum, is the most populous island in the province and home to the provincial administrative capital Isangel and the main commercial town of Lenakel. Residents from outlying islands travel to Tanna to access services unavailable on their home islands[13]. On Aneityum, an integrated malaria elimination project with mass drug administration, sustained vector control interventions, and community-directed case surveillance initiated in 1991 led to the microscopically-confirmed elimination of *P. falciparum* in 1992 and *P. vivax* in 1997[14] (Fig. 2). The community-directed case surveillance programme was conducted by volunteers, who were selected by the local health committee and trained in malaria microscopy to serve as community microscopists. These community microscopists collaborated with a registered nurse on the island to examine all arrivals (active case detection) and febrile cases (passive case detection) for *Plasmodium* infection. Community-directed case surveillance has been maintained since 1992[15].

In August 2000, a church meeting was held on Aneityum and attended by visitors from several islands in Vanuatu. Polymerase chain reaction (PCR) detected *Plasmodium* infections in 10.2% (28/274) of visitors who might have reintroduced *P. vivax* to Aneityum and triggered the outbreak in early 2002[16]. From January to March 2002, community microscopists reported 67 *P. vivax*-positive cases among 240 blood samples collected from febrile residents on Aneityum, confirming an outbreak of vivax malaria of unknown origin.

Subsequent population-wide cross-sectional community- and school-based surveys on Aneityum in July 2002 revealed *P. vivax* prevalence of 10.1% (77/759) by PCR, with 71.4% (55/77) of the infections considered as sub-microscopic (PCR-positive but microscopy-negative)[16]. Analyses of sequence polymorphisms in the genes for the surface antigens *P. vivax* merozoite surface protein-1 and *P. vivax* circumsporozoite protein from microscopically positive cases revealed fewer haplotypes in parasites from Aneityum than those from other islands. Shared haplotypes among parasites from Aneityum and other islands indicated that the parasites responsible for the outbreak in 2002 observed on Aneityum were imported. However, the exact source(s) could not be determined.

Neutral microsatellites have been used extensively in studies of the genetic diversity of *Plasmodium* populations to assess transmission levels and patterns[17,18], determine the multiplicity/complexity of infections[19,20], and infer the origin and spread of drug-resistant parasites[21,22]. Furthermore, partly owing to their high mutation rates[23], microsatellites are useful genetic markers for investigating recent malaria outbreaks[24].

We analysed *P. vivax* microsatellite markers from the 2002 Aneityum outbreak and compared them with *P. vivax* populations from earlier periods and other islands to identify the origin of the parasites responsible for the

**Fig. 1 | Map of Vanuatu.** Islands from which *P. vivax* was sampled are labelled. The inset shows the location of Vanuatu in the Southwest Pacific. All of the maps were drawn with the free software DIVA-GIS version 7.5 (http://www.diva-gis.org/).

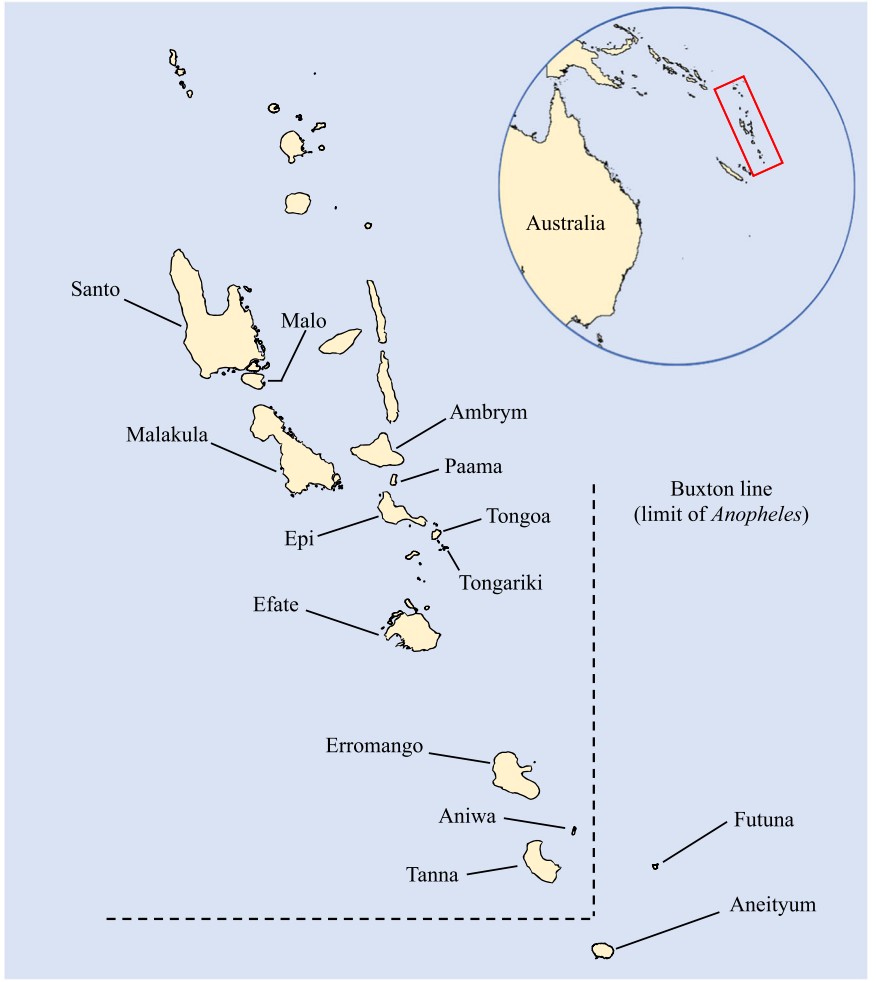

2002 outbreak. We tested the following non-mutually exclusive hypotheses: (A) the outbreak was caused by persistent *P. vivax* that had been circulating on Aneityum in 1999 and 2000; (B) the outbreak was caused by the reintroduction of *P. vivax* by church meeting attendees in 2000; and (C) the outbreak was caused by a recent *P. vivax* reintroduction from neighbouring Tanna Island in 2002.

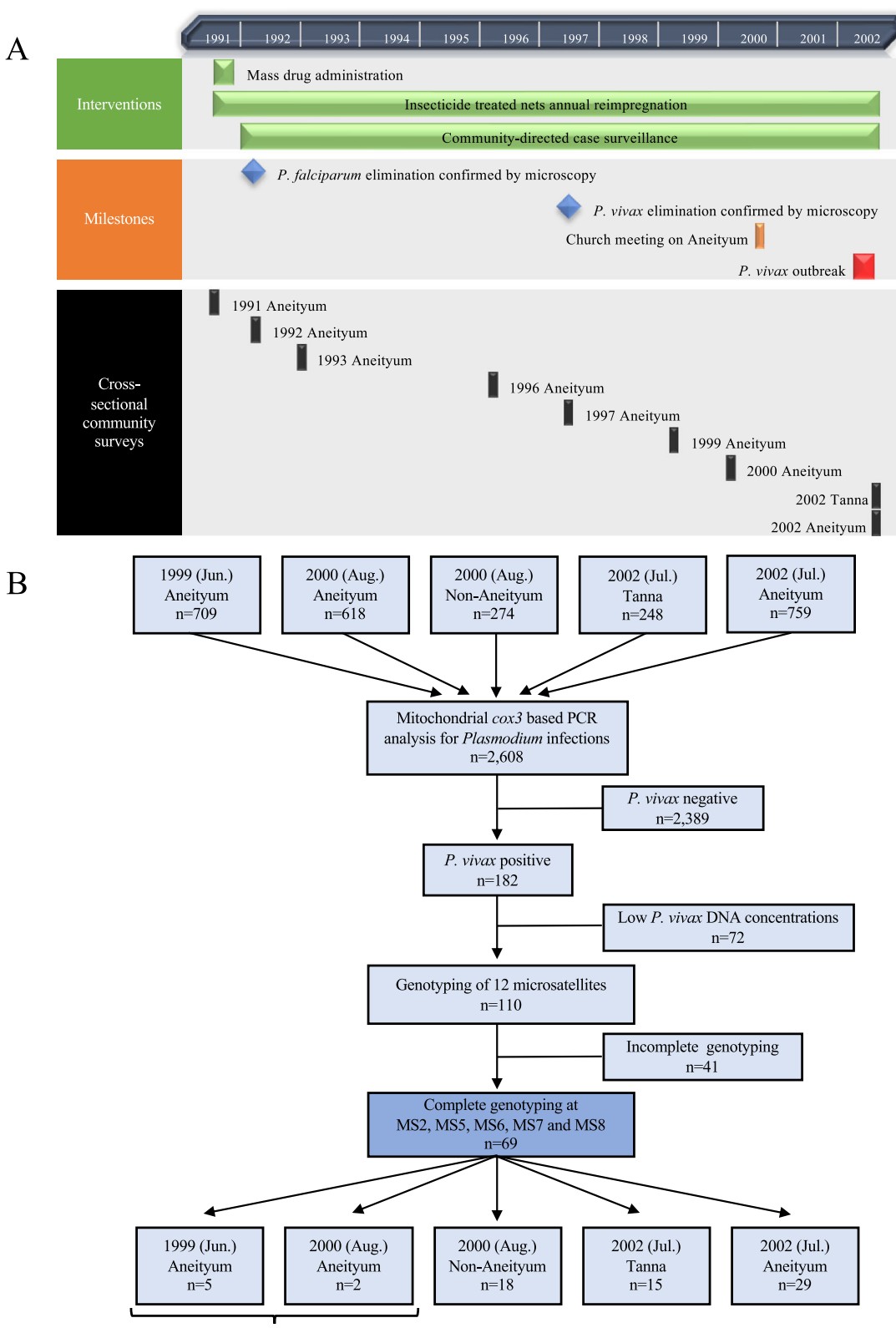

**Fig. 2 | Malaria elimination and *P. vivax* outbreak on Aneityum Island in Vanuatu and overview of sample selection stages. A** Timeline of implementation of malaria elimination interventions, major milestones and relevant events, and cross-sectional surveys on Aneityum Island from 1991 to 2002 and **B** Flowchart of *P. vivax* sample selection for microsatellite analysis

Results from this study reveal distinct genetic origins of *P. vivax* responsible for the outbreak on Aneityum. While the origin of most *P. vivax* lineages cannot be identified, limited genetic diversity among these lineages suggests a common ancestor recently introduced to the island. A recent importation of *P. vivax* from Tanna to Aneityum (hypothesis C) is implicated by the sharing of an identical microsatellite haplotype among small subsets of parasite isolates from these two islands. Phylogenetic and population structure analyses provide support for another more distant *P. vivax* importation event in 1999 or 2000 (hypotheses A and B).

## Methods

### Ethics statement

The Ethics Committee of Tokyo Women's Medical University and the Ministry of Health in Vanuatu approved the study. Ethical approval to analyse previously collected samples was provided by the Ethics Committee at Osaka Metropolitan University (approval number 3206). Oral informed consent was obtained from all adult participants and parents in the case of children due to the lack of formal writing systems for local 'kastom' languages in Vanuatu. The study's purpose, procedure, and potential risks and benefits were explained to participants in Bislama (lingua franca understood by most adults in Vanuatu) and local languages by ni-Vanuatu collaborators. The consent was witnessed by a neutral third party (e.g. teacher, village chief) who recorded the name of each participant at study enrolment. All study procedures were conducted in adherence to the Declaration of Helsinki.

### Sample collection and detection of *Plasmodium* infections

Cross-sectional, community- and school-based malariometric surveys were conducted on Aneityum Island in June 1999, August 2000, July 2002, and Tanna Island in July 2002. In 2000, the survey on Aneityum included 274 individuals from other islands (Santo, Malo, Ambrym, Malakula, Paama, Tongoa, Tongariki, Epi, Efate, Erromango, Tanna, Aniwa, and Futuna) who had come to Aneityum to attend a church meeting (August 23-28, 2000). *P. vivax* populations are henceforth denoted by the year and island origin (e.g., 1999 Aneityum). *P. vivax* isolates detected among church meeting attendees in 2000 are collectively referred to as 2000 non-Aneityum.

Demographic information of the survey participants, including age, sex, and village of residence, was recorded. Using microscopy and PCR, capillary blood samples were collected via finger pricking to examine *Plasmodium* infections. For microscopic examination, the thin blood film was fixed with methanol, and both thick and thin films were stained with 10% Giemsa solution for 10 min. An experienced microscopist examined the stained blood films. Two additional blood samples (70 μL each) were collected as dried blood spots (DBS) on Whatman ET31 Chr filter paper (Whatman International, Maidstone, UK) using a 75-mm heparinised haematocrit capillary tube (Thermo Fisher Scientific, MA, USA). After drying, DBS was stored in individual plastic bags at ambient temperature in the field and then at −20 °C upon arrival at the laboratory in Japan. DNA was extracted from a quartered DBS (corresponding to 17.5 μL of blood) using the QIAamp Blood Mini Kit (Qiagen, Hilden, Germany) according to the manufacturer's protocol. *Plasmodium* infections were detected using the nested PCR method that targets the mitochondrial cytochrome c oxidase-3 (*cox3*) gene, using 4 μL of extracted DNA (equivalent to 0.47 μL of blood) as template[25].

### *P. vivax* nuclear DNA quality verification

To ensure the presence and quality of *P. vivax* nuclear DNA, all mitochondrial *cox3* PCR-positive samples were subject to nested PCR amplifications of three nuclear loci, *msp1F1*, *msp1F3* and MS16[26]. Each 10-μL reaction consisted of 5 μL of the GoTaq Green Master Mix (Promega, WI, USA) and 1.25 μM of each primer (Supplementary Data 1). We used 3 μL of DNA (equivalent to 0.35 μL of blood) as a template for the primary PCR and 1 μL of the primary PCR product for the nested PCR. The cycling conditions for the primary PCR were as follows: 95 °C for 1 min; 40 cycles at 95 °C for 15 s, 59 °C for 30 s, and 72 °C for 30 s, and 72 °C for 30 s, and a final

extension at 72 °C for 5 min. The cycling conditions for the nested PCR were identical to those of the primary PCR, except that only 30 cycles were performed. The presence of PCR amplicons after the nested round was detected using the LabChip GX Touch (PerkinElmer, MA, USA). Samples that failed to amplify two or more quality verification nuclear markers were excluded from microsatellite analysis.

### Genotyping of microsatellite markers

Samples positive for two or more quality verification nuclear markers were used for microsatellite analysis. Twelve validated highly polymorphic microsatellite markers (MS1, MS2, MS5, MS6, MS7, MS8, MS9, MS10, MS12, MS15, MS20 and MS3.27) were amplified using hemi-nested PCR[27]. Each 10-μL reaction consisted of 5 μL of the PrimeSTAR Max Premix (Takara-Bio, Kusatsu, Japan) and 1.25 μM of each primer (Supplementary Data 1). The primary PCR used 3 μL of DNA as a template, while the nested PCR used 1 μL of the primary PCR product. The cycling conditions for the primary PCR were as follows: 95 °C for 1 min; 40 cycles at 95 °C for 15 s, 60 °C for 30 s, and 72 °C for 30 s, and a final extension at 72 °C for 5 min. The cycling conditions for the nested PCR were as follows: 95 °C for 1 min, 35 cycles at 95 °C for 15 s, 61 °C for 30 s, 72 °C for 45 s and a final extension at 72 °C for 5 min. The nested PCR products were resolved on an Applied Biosystems 3730xl DNA Analyzer using the GeneScan 600 LIZ size standard (Thermo Fisher Scientific). Allele sizes were determined using Peak Scanner 2 (Thermo Fisher Scientific). Alleles with a minimum of one-third of the relative fluorescence units of the primary allele were recorded as secondary alleles.

### Population genetic analysis

Because most samples (68.2%; 75/110) included in the microsatellite analysis were sub-microscopic and likely contained limited *P. vivax* DNA, PCR amplification success rates varied among markers despite multiple attempts (Supplementary Data 2). To include as many samples as possible in subsequent analyses, five markers (MS2, MS5, MS6, MS7 and MS8) with the most genotyped samples were included. The microsatellite haplotype was constructed for each sample using only the primary allele at each marker.

The genetic diversity of each microsatellite locus was assessed by calculating the number of alleles and expected heterozygosity ($H_E$) in each population. In addition, haplotype diversity ($h$) was used to assess *P. vivax* population diversity. $H_E$ and $h$ were calculated using the following equations:

$$H_E \text{ or } h = \frac{n}{n-1}\left(1 - \sum_1^n P_i^2\right) \quad (1)$$

Where *n* represents the number of analysed samples, and $P_i$ represents the frequency of the ith allele or ith haplotype in the population[28]. The mean number of allele differences between pairs of haplotypes in each population was calculated using Arlequin 3.5.2.2[29].

Multilocus linkage disequilibrium (LD) among microsatellites was assessed using LIAN 3.7[30]. The standardised index of association ($I^S_A$) was calculated as follows:

$$I^S_A = \frac{1}{n-1}\left(\frac{V_D}{V_e} - 1\right) \quad (2)$$

Where n is the number of loci analysed, $V_D$ is the observed variance in the number of loci differing between each pair of haplotypes, and $V_e$ is the expected variance under the linkage equilibrium. To investigate whether the observed data differed from random expectations, we compared $V_D$ values with the distribution of $V_e$ values using Monte Carlo simulations with 10,000 random resamplings of simulated datasets, in which alleles at each locus were randomly reshuffled among genotypes.

Genetic differentiation among populations was assessed using pairwise $F_{ST}$ genetic distances, as described in Arlequin 3.5.2.2[29]. The statistical significance of the $F_{ST}$ genetic distances was evaluated by randomly permuting

haplotypes between sites approximately 10,000 times to generate a null distribution against which the observed values were compared.

To determine relationships among *P. vivax* microsatellite haplotypes, Populations 1.2.32[31] was used to calculate Nei's $D_A$ genetic distance[28,32] and MEGA11[33] was used to construct an unrooted neighbour-joining tree. To detect the *P. vivax* population genetic structure, we used the admixture model[34] with island as prior information (LOCPRIOR) in STRUCTURE version 2.3.4[35]. The number of genetic clusters (K) was set from 1 to 10 and estimated using a Markov Chain Monte Carlo algorithm with 50,000 burn-in and 50,000 iterations per replication over 20 replications for each K. Using STRUCTURE HARVESTER Web v0.6.94[36], we calculated ΔK[37], the rate of change in the log probability between successive Ks, to estimate the optimal K. Principal component analysis (PCA) was used as a non-model-based alternative to STRUCTURE to infer the *P. vivax* population genetic structure. PCA was performed using the Adegenet package[38] in R version 4.2.1.

### Reporting summary

Further information on research design is available in the Nature Portfolio Reporting Summary linked to this article.

## Results

### Detection of *Plasmodium* infections using microscopy and PCR

Of 2608 samples available from the 1999–2002 period in Vanuatu (Fig. 2), PCR detected 219 (8.4%) *Plasmodium* infections, of which 57 (2.8%) were also detected by microscopy. *P. vivax* accounted for most microscopic (75.4%; 43/57) and sub-microscopic (85.8%; 139/162) infections. Throughout the study period, neither microscopy nor PCR detected any species other than *P. vivax* on Aneityum Island, and the prevalence of *P. vivax* infection by PCR increased from 0.8% (6/709) in 1999 to 10.1% (77/759) in 2002 (Table 1).

### Selection of *P. vivax* samples for microsatellite analysis

PCR amplification of at least two nuclear loci used for DNA quality verification failed in 72 of the 182 mitochondrial *cox3* PCR-positive *P. vivax* samples. Of the remaining 110 samples subject to PCR amplification of 12 microsatellites, complete genotyping at five markers (MS2, MS5, MS6, MS7 and MS8) was successful in 69 samples (24 microscopic and 45 sub-microscopic): five from 1999 Aneityum, two from 2000 Aneityum, 18 from 2000 non-Aneityum, 15 from 2002 Tanna and 29 from 2002 Aneityum. Due to small sample sizes, the 1999 and 2000 Aneityum samples were combined as 1999–2000 Aneityum and analysed as a single 'pre-outbreak' population (Fig. 2).

### Microsatellite diversity in *P. vivax* populations

Unbiased $H_E$ of the five microsatellite markers in each population is summarised in Table 2. Except for MS8, $H_E$ was consistently lower in the 2002 Tanna and 2002 Aneityum populations. Limited polymorphisms were observed in MS5 ($H_E \pm SE$: 0.229 ± 0.003) and MS6 (0.254 ± 0.004) in 2002 Aneityum, and MS7 (0.342 ± 0.028) in 2002 Tanna.

Fifty-three unique microsatellite haplotypes were identified in 69 samples (Supplementary Data 3). In the 1999–2000 Aneityum and 2000 non-Aneityum samples, each *P. vivax* isolate harboured a unique haplotype (Table 2). In 2002 Tanna, 12 unique haplotypes were identified among 15 samples ($h \pm SE$: 0.971 ± 0.011), and three haplotypes (26, 28 and 29) were each shared between two samples. In 2002 Aneityum, 17 unique haplotypes were identified in 29 samples (0.836 ± 0.027). Haplotype 39 was found in 41.4% (12/29) of the samples, whereas another closely related haplotype 47 was shared by two samples. Haplotype 26 was shared between two 2002 Tanna and one 2002 Aneityum samples. Haplotype 42, which differed from haplotype 26 at MS8, was found in another 2002 Aneityum sample (Supplementary Data 3).

The mean number of allele differences between the pairs of samples in each population was calculated as another measure of within-population genetic variation (Table 3). The mean was lower in the 2002 Tanna

(mean ± SD: 2.857 ± 1.267) and 2002 Aneityum (2.091 ± 1.562) populations than in the 1999–2000 Aneityum (4.238 ± 0.865) and the 2000 non-Aneityum (4.562 ± 0.538) populations. Consistent with this finding, significant LD among loci was observed only in 2002 Tanna ($p = 2.6 \times 10^{-3}$) and 2002 Aneityum ($p < 2.0 \times 10^{-5}$) (Table 4).

### Genetic relatedness and structuring of *P. vivax* populations

$F_{ST}$ analyses of pairwise genetic distances indicated that the 2002 Aneityum and the 2002 Tanna populations differed significantly from each other and other populations ($p < 0.05$) (Table 3). However, no statistically significant difference was observed between the 1999–2000 Aneityum population and the 2000 non-Aneityum population and between microscopic ($n = 13$) and sub-microscopic ($n = 16$) infections within the 2002 Aneityum population ($F_{ST} = 0.001$; $p = 0.42$).

Two major clades were identified in the unrooted neighbour-joining tree. The first comprised all 15 isolates from 2002 Tanna, two from 2002 Aneityum (haplotypes 26 and 42), and one from 2000 non-Aneityum. The second consisted of six of seven isolates from 1999–2000 Aneityum and 16 of 18 isolates from 2000 non-Aneityum (Fig. 3A). Both STRUCTURE and PCA revealed population structure among the four *P. vivax* populations. Bayesian clustering assignments revealed that the most likely number of clusters (K) was three (Fig. 3B, C). Two lineages (haplotypes 26 and 42) in the 2002 Aneityum population exhibited patterns of inferred ancestry similar to those in the 2002 Tanna population. PCA also revealed a similar pattern, with three distinct clusters; the first consisted exclusively of lineages from 2002 Aneityum, the second consisted of lineages from 2002 Tanna, and two lineages from 2002 Aneityum (haplotypes 26 and 42), and the third consisted of haplotypes from 1999 to 2000 Aneityum and 2000 non-Aneityum (Fig. 3C).

## Discussion

In this study, we analysed the genetic variation at five microsatellite markers from microscopic and sub-microscopic *P. vivax* infections using samples collected from Aneityum and other islands before and during the 2002 Aneityum outbreak. Our results indicated that most *P. vivax* lineages identified during the outbreak were genetically distinct from those on Aneityum in 1999 and 2000 (hypothesis A) and among church meeting attendees in 2000 (hypothesis B). However, a small number of *P. vivax* isolates during the outbreak appeared to have originated from Tanna Island (hypothesis C).

Infrastructure was (and remains) limited on Aneityum, prompting many residents to travel to Tanna to access services including secondary education and hospital care. The observation that *P. vivax* isolates from Aneityum and Tanna shared an identical microsatellite haplotype 26 recapitulated the importance of human-mediated parasite movement among islands in Vanuatu[39,40] and highlighted the ease with which an outbreak could be seeded. Although we could not definitively ascertain the direction of *P. vivax* gene flow, we noted that both haplotype 26 and the closely related haplotype 42 were nested within the 2002 Tanna clade in the unrooted neighbour-joining tree (Fig. 3A) and clustered with the Tanna lineages instead of the remaining Aneityum lineages in PCA (Fig. 3C). Phylogenetic analysis (Fig. 3A) and PCA (Fig. 3C) also indicated genetic affinity among one 2002 Aneityum sample and one 1999–2000 Aneityum and one 2000 non-Aneityum samples. The 1999–2000 Aneityum sample was derived from a male child who had moved to Aneityum from Pentecost. The child was diagnosed with *P. vivax* in May 1999 and again in July 1999 during our survey. The 2000 non-Aneityum sample was derived from a male adult church meeting attendee from Tanna. Not only did these infections exemplify the vulnerability of Aneityum to *P. vivax* importation, but they also suggested the possibility that one of the seeds for the 2002 outbreak may have been introduced in 1999 or 2000. Notably, we previously indicated the potential of *P. vivax* relapses occurring up to five years after the initial infection[16], raising the possibility that the 2002 outbreak was partially seeded from long-latency relapses from infections initially acquired years before.

**Table 1 | *Plasmodium* infections detected by microscopy and PCR, Vanuatu, 1999–2002**

| Year | Province | Island | Number of survey participants | Mean age ± SD | Sex | | Micro-scopy[c] | | PCR only | | | | Number of *Pv*genotyped[b] |
|------|----------|--------|------|------|------|------|------|------|------|------|------|------|------|
| | | | | | Male | Female | *Pv* | *Pf* | *Pv* | *Pf* | *Pv + Pf* | *Pm* | |
| 1999 | Tafea | Aneityum | 709 | 41.6 ± 10.5 | 351 | 358 | 1[a] | 0 | 5 | 0 | 0 | 0 | 5 |
| 2000 | Tafea | Aneityum | 618 | 48.9 ± 9.0 | 329 | 289 | 4 | 0 | 2 | 0 | 0 | 0 | 2 |
| | | Futuna | 17 | 49.3 ± 19.4 | 7 | 10 | 0 | 0 | 0 | 0 | 0 | 0 | 0 |
| | | Aniwa | 3 | 28.0 ± 0.7 | 1 | 2 | 0 | 0 | 0 | 0 | 0 | 0 | 0 |
| | | Tanna | 49 | 32.2 ± 15.0 | 29 | 20 | 1 | 0 | 1 | 0 | 1 | 0 | 4 |
| | | Erromango | 4 | 47.0 ± 17.5 | 2 | 2 | 0 | 0 | 0 | 0 | 0 | 0 | 0 |
| | Shefa | Epi | 15 | 41.9 ± 6.2 | 8 | 7 | 0 | 0 | 1 | 3 | 0 | 0 | 0 |
| | | Tongoa | 2 | 45.5 | 2 | 0 | 0 | 0 | 0 | 0 | 0 | 0 | 0 |
| | | Tongariki | 1 | 32 | 1 | 0 | 0 | 0 | 0 | 0 | 0 | 0 | 0 |
| | | Efate | 91 | 42.2 ± 13.8 | 53 | 38 | 0 | 0 | 6 | 0 | 0 | 0 | 6 |
| | Malampa | Ambrym | 17 | 37.4 ± 9.7 | 10 | 7 | 0 | 1 | 2 | 0 | 0 | 0 | 2 |
| | | Malakula | 46 | 36.9 ± 10.1 | 26 | 20 | 0 | 0 | 7 | 1 | 1 | 1 | 4 |
| | | Paama | 5 | 38.8 ± 13.0 | 3 | 2 | 0 | 0 | 0 | 0 | 0 | 0 | 0 |
| | Sanma | Santo | 23 | 39.6 ± 8.1 | 12 | 11 | 1 | 1 | 1 | 0 | 0 | 0 | 2 |
| | | Malo | 1 | 39 | 1 | 0 | 0 | 0 | 0 | 0 | 0 | 0 | 0 |
| 2002 | Tafea | Tanna | 248 | 15.0 ± 13.3 | 110 | 138 | 14 | 12 | 59 | 6 | 10 | 0 | 15 |
| | | Aneityum | 759 | 21.1 ± 17.9 | 386 | 373 | 22 | 0 | 55 | 0 | 0 | 0 | 29 |
| | | Total | 2608 | | 1331 | 1277 | 43 | 14 | 139 | 10 | 12 | 1 | 69 |

*Pv P. vivax*, *Pf P. falciparum*, *Pm P. malariae*, SD standard deviation.
[a]Moved from Pentecost.
[b]Completely genotyped for MS2, MS5, MS6, MS7, and MS8.
[c]All microscopy-positive infections are PCR-positive.

**Table 2 | Unbiased expected heterozygosity ($H_E$) of five *P. vivax* microsatellite markers and haplotype diversity (*h*) in four populations from Vanuatu**

| | Unbiased expected heterozygosity ($H_E$) ± SE | | | | | Haplotype diversity (*h*) ± SE | |
|------|------|------|------|------|------|------|------|
| | MS2 | MS5 | MS6 | MS7 | MS8 | $N_H$ | *h* |
| 1999–2000 Aneityum (*n* = 7) | 0.978 ± 0.011 | 0.889 ± 0.011 | 0.889 ± 0.017 | 0.873 ± 0.013 | 0.800 ± 0.021 | 7 | 1.000 |
| 2000 non-Aneityum (*n* = 18) | 0.976 ± 0.002 | 0.879 ± 0.003 | 0.948 ± 0.002 | 0.967 ± 0.002 | 0.628 ± 0.009 | 18 | 1.000 |
| 2002 Tanna (*n* = 15) | 0.822 ± 0.015 | 0.537 ± 0.024 | 0.630 ± 0.025 | 0.342 ± 0.028 | 0.693 ± 0.020 | 12 | 0.971 ± 0.011 |
| 2002 Aneityum (*n* = 29) | 0.517 ± 0.003 | 0.229 ± 0.003 | 0.254 ± 0.004 | 0.483 ± 0.003 | 0.707 ± 0.005 | 17 | 0.836 ± 0.027 |

$N_H$ number of unique haplotypes, *SE* standard error.

The 2002 Tanna samples included in this study were collected through cross-sectional community surveys in early July, approximately a week before the cross-sectional surveys on Aneityum that supplied the *P. vivax* outbreak samples. These observations suggested that the importation of *P. vivax* from Tanna likely had a direct, albeit minor, contribution to the outbreak in Aneityum.

Notably, the 2002 Tanna population shared many characteristics with the 2002 Aneityum population, including lower expected heterozygosity and haplotype diversity (Table 2), lower mean pairwise differences (Table 3), and significant LD among loci (Table 4). Nationally, the increase in malaria incidence during the early 2000s was attributed to the combined effect of an earthquake followed by a tsunami and a general reduction in funding for the malaria programme[13,41]. Lower *P. vivax* genetic diversity in the 2002 Tanna population was consistent with heightened transmission and rapid expansion of a limited number of parasite clones.

Although the geographic origin of most *P. vivax* samples on Aneityum in 2002 remains elusive, our findings suggested that this population is semi-clonal, with 41.4% (12/29) of the isolates represented by a single microsatellite haplotype (Supplementary Data 3). The samples included in this study were collected in July 2002, a few months after the onset of the outbreak reported by community microscopists in January[16] and the typical peak transmission season between February and March[12]. While the mutation rates of microsatellites and recombination rates in *P. vivax* are unknown[42,43], the limited microsatellite diversity observed in this population suggested that most *P. vivax* lineages on Aneityum in 2002 likely descended from a small number of recent common ancestors.

In our previous study, we speculated that visitors who attended the church meeting in 2000 were a potential source of the parasites responsible for the 2002 outbreak in Aneityum[16]. However, our findings from the current study did not support a direct causal link between *P. vivax* lineages among visitors and those among Aneityum residents 2 years later, as no

**Table 3 | Average pairwise differences (± standard deviation) within population (diagonal) and pairwise $F_{ST}$ genetic distances (lower triangle) among four _P. vivax_ populations in Vanuatu**

| | 1999–2000 Aneityum | 2000 non-Aneityum | 2002 Tanna | 2002 Aneityum |
|---|---|---|---|---|
| 1999–2000 Aneityum | 4.238 ± 0.865 | | | |
| 2000 non-Aneityum | 0.022 | 4.562 ± 0.538 | | |
| 2002 Tanna | 0.280* | 0.213* | 2.857 ± 1.267 | |
| 2002 Aneityum | 0.398* | 0.324* | 0.488* | 2.091 ± 1.562 |

Genetic differentiation inferred from significant ($p < 0.05$) $F_{ST}$ distances is indicated by an asterisk (*).

**Table 4 | Linkage disequilibrium among five microsatellite markers in four _P. vivax_ populations**

| | $V_D$ | $V_e$ | $I^s_A$ | Monte Carlo simulation | | |
|---|---|---|---|---|---|---|
| | | | | Var ($V_D$) | p value | L |
| 1999–2000 Aneityum | 0.6905 | 0.6122 | 0.0319 | 0.0332 | 0.26 | 0.9905 |
| 2000 non-Aneityum | 0.3399 | 0.3471 | −0.0052 | 0.0011 | 0.66 | 0.4057 |
| 2002 Tanna | 1.6044 | 1.0824 | 0.1205 | 0.0205 | $2.6 \times 10^{-3}$ | 1.3544 |
| 2002 Aneityum | 2.4386 | 1.0618 | 0.3242 | 0.0408 | $<2.0 \times 10^{-5}$ | 1.4262 |

$V_D$: Observed mismatch variance.
$V_e$: Mismatch variance expected under linkage equilibrium.
$I^s_A$: Standardised index of association.
Var ($V_D$): Variance of $V_D$.
L: 5% critical value for $V_D$.

shared haplotypes were observed between the 2000 non-Aneityum and 2002 Aneityum _P. vivax_ populations. $F_{ST}$-based analysis of genetic differentiation (Table 3) and population structure using STRUCTURE and PCA (Fig. 3B, C) indicated that the two _P. vivax_ populations were distinct.

We previously identified microscopic _P. vivax_ infections only in individuals aged ≤20 (born after 1981) and concluded that persisting anti-_P. vivax_ IgG antibodies among adults routinely exposed to _P. vivax_ before elimination may have limited the infections to sub-microscopic levels in this cohort[16]. In this study, we did not observe significant differentiation ($F_{ST} = 0.001$; $p = 0.42$) between the lineages of microscopic and sub-microscopic infections in Aneityum in 2002. Therefore, additional microsatellite markers and the genotyping or sequencing of other host immunity-targeted _P. vivax_ antigen genes are required to assess whether distinct parasitic clones cause microscopic and sub-microscopic infections.

Our study had several limitations. First, the sample quality and availability limited our ability to genetically characterise all available _P. vivax_ samples. Most samples in this study were obtained from asymptomatic and sub-microscopic _P. vivax_ infections with low parasitaemia levels, which limited the number of samples and microsatellites included in our analyses. Of the 182 _P. vivax_-positive samples by the mitochondrial _cox3_ PCR, 49 (26.9%) failed to amplify any nuclear loci, while 23 (12.6%) amplified only one nuclear locus. These 72 samples were deemed to have low DNA concentrations and were excluded from microsatellite marker analysis (Fig. 2). Molecular diagnostic yields are mainly determined by blood volume analysed and the gene copy number of the amplified molecular marker[44]. The discrepancy in PCR amplification efficiencies in our study likely reflected such a difference in gene copy numbers between the mitochondrial _cox3_ and

the nuclear microsatellite markers. Within each parasite, there are 20 to 150 copies of the mitochondrial _cox3_ gene[45], but only one copy of nuclear microsatellite markers, making the former an ideal target for PCR detection of _Plasmodium_ infections. Low _P. vivax_ nuclear DNA concentrations in most samples, and a lack of availability of advanced methods and analytical approaches, limited our ability to gain a more comprehensive understanding of the population genomic diversity of _P. vivax_ parasites. Recent developments in selective whole genome amplification (sWGA) of _Plasmodium_ species from samples stored as DBS and in amplicon deep sequencing have provided opportunities to include low-parasite density infections in malaria genomic epidemiology[46–50]. Nevertheless, there is some discussion that advanced malaria genomics may have several gaps for an immediate "real-world malaria control and elimination strategy"[51] and may not be necessary for malaria control or even its elimination[52]. Second, _P. vivax_ samples from other islands in Vanuatu in 2000 were collected from church meeting attendees and thus were not representative of the genetic diversity of the parasites on those islands. Third, 2002 Aneityum samples were collected six months after the onset of the outbreak. Additional but undetected parasite reintroduction events between the onset of the outbreak in early 2002 and our survey in July 2002 could have obscured or even replaced the original _P. vivax_ clone that triggered the outbreak. Lastly, the primary objective of the cross-sectional survey on Aneityum in July 2002 was to determine _P. vivax_ prevalence in the communities. As it was not an outbreak investigation, we did not enquire about participants' travel histories or distinguish whether an active infection resulted from a recent inoculation by an infected _Anopheles_ vector or a relapse from activation of latent hypnozoites. In endemic countries, a substantial proportion of _P. vivax_ cases is reportedly caused by the activation of hypnozoites with genotypes distinct from those that caused the initial infections[53]. Detecting potential relapses from heterologous hypnozoites among the _P. vivax_ infections in the 2002 Aneityum samples could have complicated the utilisation of genotyping or DNA sequencing to determine the source of the parasites responsible for the outbreak. Since _P. vivax_ infections in adults were sub-microscopic due to persisting immunity acquired before malaria elimination and undetectable by diagnostics available locally, coupled with the potential of long-latency relapse, the 2002 outbreak's seeding event could have occurred years ago, and the outbreak itself could have been triggered by a recent relapse. Determining the proportion of infections resulting from relapse during _P. vivax_ outbreaks can inform the most optimal response strategy, however distinguishing between primary infections and relapses in settings such as Aneityum or Vanuatu, where conditions conducive to transmission remain, is difficult.

Although we lack entomological data, the observation that most _P. vivax_ isolates during the 2002 outbreak were genetically very similar suggests that transmission by _Anopheles_ mosquitoes played a major role in sustaining the outbreak. Therefore, it can be inferred that the level of receptivity was probably very high in 2002. In response, a second round of mass drug administration was implemented, targeting individuals under 20 years of age. At the same time, provisions of insecticide-treated nets (ITNs) were strengthened, coupled with a robust malaria health promotion programme to raise awareness of the continuous risk of malaria resurgence among local communities[15]. Subsequent annual cross-sectional surveys revealed low prevalence (1.9 to 3.9%) of _P. vivax_ infections by PCR between 2003 and 2007. Since 2010, no _Plasmodium_ infections have been detected by microscopy and PCR[54]. Nonetheless, 74.1 and 67.4% of the island's residents reported ITN use in 2016 and 2019, respectively (unpublished data). We believe that high ITN usage likely suppressed malaria receptivity on Aneityum Island.

Vanuatu aims to interrupt malaria transmission and achieve zero indigenous malaria cases nationally by the end of 2023 through a multi-stage process in which elimination is targeted first in the most peripheral provinces with low incidences (Tafea, Torba and Penama), followed by central provinces with intermediate (Shefa) and finally high incidences (Malampa and Sanma)[13]. Key to this strategy is preventing the reintroduction of _Plasmodium_ parasites and re-establishing their transmission in provinces

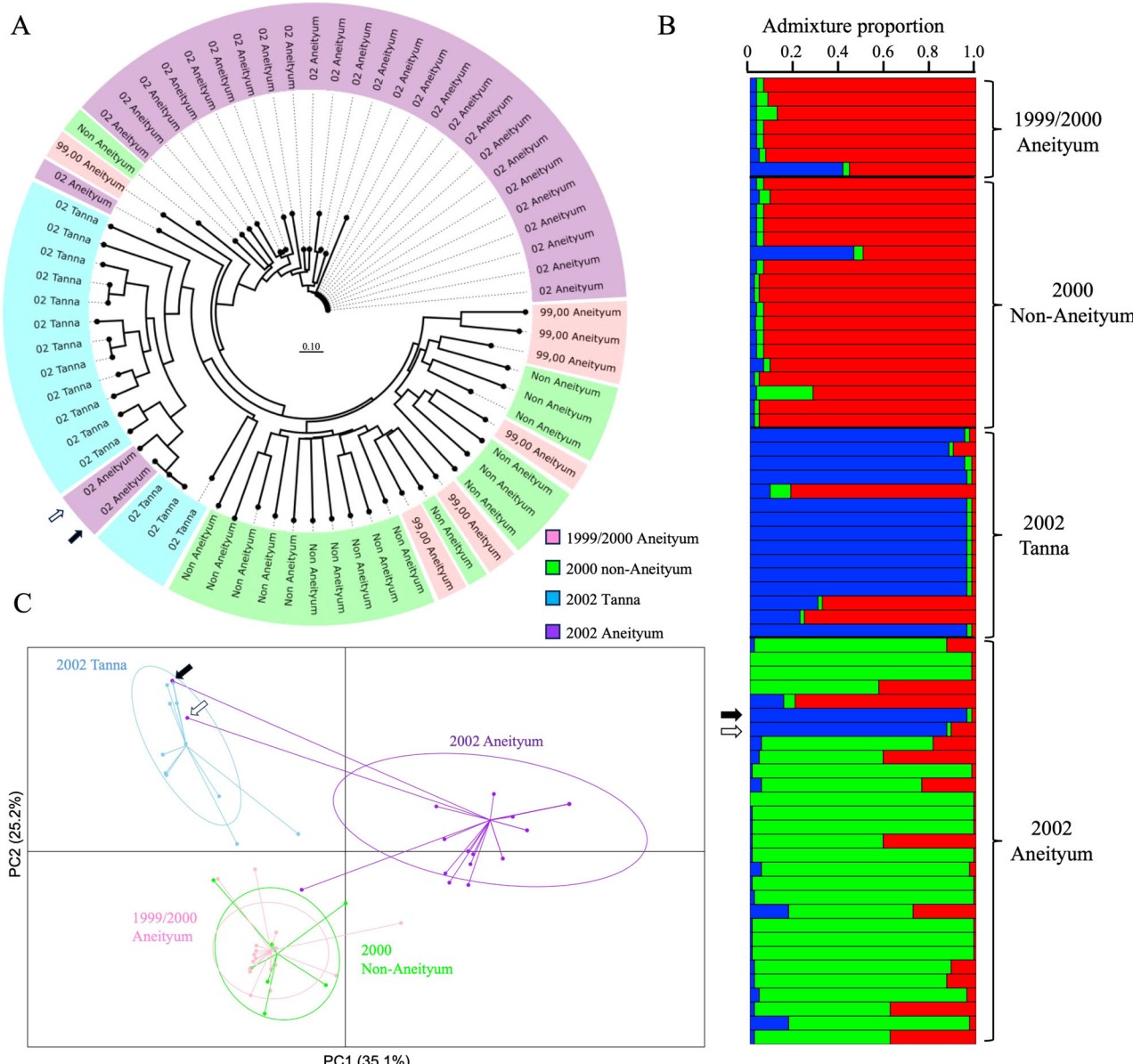

**Fig. 3 | Genetic relatedness and structure inferred from five nuclear microsatellite markers (MS2, MS5, MS6, MS7 and MS8) among 69 *P. vivax* isolates from four populations in Vanuatu, 1999–2002. A** An unrooted neighbour-joining tree based on Nei's D$_A$ genetic distances. **B** Inference of population structure using the admixture model in STRUCTURE. Each bar represents an individual *P. vivax* isolate. The isolates are grouped by populations identified by year and geographical origin. The most likely number of ancestral populations (K) is three, each represented by a colour. **C** Principal component analysis (PCA) plot of *P. vivax* microsatellite haplotypes. Arrows indicate *P. vivax* isolates of likely Tanna origin in the 2002 Aneityum outbreak.

where malaria has been successfully eliminated. Case-based surveillance and rapid response are imperative to sustaining malaria elimination at the subnational level. Despite surveillance among arrivals and suspected febrile cases in communities, high coverage of ITNs, and persisting antimalarial immunity among residents, an outbreak of *P. vivax* was reported on Aneityum only five years after the parasite had been eliminated, highlighting both the high receptivity and high vulnerability of islands in Vanuatu to resurgence[15,16]. Genetic epidemiological methods have provided fruitful insights in many epidemiological investigations. However, our findings suggest that pinpointing the origins of parasites among many potential sources during a reintroduction event based on genetic or genomic methods alone can be challenging.

## Data availability
Source data used for results and figures are provided in the Supplementary Data 4. The authors declare that all data supporting the findings of this study are available within the article and its Supplementary Information files are available from the authors upon request.

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

## Acknowledgements
We want to express our sincere gratitude to the participants in this study. We are also grateful to Mayumi Fukui, Ikuko Kusuda, Satoko Naito, Tomomi Kuwana and all the supporting staff in the Ministry of Health in Vanuatu for their assistance. This project was supported by grants from the Swedish Research Council (2008-3097; 2009-3233 and 2022-04055), Formas (2022-02360), the Japanese Society for the Promotion of Science (24390141; 22406008; Asia-Africa Science Platforms; 21K02817), Health Labour Sciences and the Global COE Programme at Nagasaki University.

## Author contributions
C.W.C. and Akira Kaneko designed the study. M.K., S.Y., H.I. and Akira Kaneko conducted surveys and obtained samples used in this study. S.S. and C.W.C. performed microsatellite genotyping data analyses. S.S. and C.W.C. wrote the first draft of the manuscript, and S.S., C.W.C., M.K., S.Y., H.I., G.T., A.KC., W.K., Y.K. and Akira Kaneko reviewed, revised and approved the final manuscript.

## Funding

## Competing interests
The authors declare no competing interests.
