## [Peer Review File · Communications Medicine]

Reviewers' comments:

Reviewer #1 (Remarks to the Author):

The manuscript by Sekine et al entitled "A quest for the origins of Plasmodium vivax resurgence after the successful malaria elimination on Aneityum Island, Vanuatu" that aimed to describe the origin of an P. vivax outbreak in 2002 at Aneityum Island by comparing microsatellite markers from outbreak with earlier periods and other islands. Findings suggest that the outbreak population is semi-clonal and that a small number of P. vivax isolates during the outbreak may have originated from Tanna Island. However, the origin remains elusive. As authors mention in the discussion, this study shows that pinpointing the origins of parasites among many potential sources during a reintroduction event can be challenging. Main limitations, clearly exposed by authors, are the limited number of samples due to low densities, the potential relapses from heterologous hypnozoites among the P. vivax infections in the 2002 Aneityum samples, and possible undetected parasite reintroduction events during the intervening period.

The manuscript is very well written, with clear hypothesis and methods. I only have minor comments:

- PCR amplification of at least two nuclear loci used for DNA quality verification failed in 72 of the 182 mitochondrial cox3 PCR-positive P. vivax samples: does this raise any concern about the performance (false positives?) of the cox3 PCR?
- Adding a dispersion measure of the mean number of allele differences between paired samples would make more robust the claims of lower mean in the 2002 Tanna and 2002 Aneityum populations than in the 1999-2000 Aneityum and the 2000 non-Aneityum populations.
- Check line 45 (some letters are missing)

Reviewer #2 (Remarks to the Author):

The article presented by the authors offers a compelling case study of a malaria outbreak on an island long after the disease had been eliminated from that location. The authors have clearly described their aim to investigate the origins of the malaria parasites responsible for the 2002 outbreak in Anyeityum, Vanuatu, alongside their hypotheses. Given the increasing trend of malaria elimination across many countries, it is indeed crucial to grasp how we can sustain these successes, making this topic particularly relevant and pressing.

While I understand that some aspects of the data and the context of this outbreak have been explored in previous publications, this paper provides a fresh lens by incorporating genetic analysis and additional data from a neighbouring island. I'd like to note that while my expertise does not extend to the nuances of genetic analysis and the specific methods applied in this manuscript, I am keen to offer my comments on the epidemiological aspects of the paper:

- In Figure 3, I noticed an intriguing observation where a sample from the 2002 Aneityum samples closely aligns with the 1999-2000 Aneityum and 2000 non-Aneityum clusters. This particular observation doesn't seem to be emphasised in the manuscript. Do the authors consider the proximity of this sample to be insignificant? If so, elaborating on the reasons might be important,

given this observation might explain the connections between the 2002 and 2000 populations.

- While the authors have underscored the significance of relapses in *P. vivax* control in the introduction, there is limited discussion on the potential role of relapses in seeding the 2002 outbreak. The 2000 study [1] indicated that relapses could be observed up to 5 years after the initial infection. Could these long-latency relapses from prior infections have been a contributing factor to the onset of the outbreak?
- Taking into account the immunity resulting from previous infections (predominantly in adults), the frequency of travel (primarily by adults, I assume), which together might result in importation of asymptomatic cases, and coupled with the potential long-latency of relapse: How probable is it, in the authors' opinion, that the outbreak's seeding occurred in the past, possibly linked to travels beyond Tanna (explaining the minimal connection with Tanna samples) and was potentially triggered by a recent relapse?
- What is the level of receptivity of Aneityum to malaria transmission? An outbreak resulting in 10% of the population being infected at its tail end suggests that the island remains highly receptive to transmission. Were there subsequent outbreaks in the following years? It's curious that while the introduction of cases seemed to occur somewhat frequently, and the island appeared highly receptive to transmission, actual outbreaks were comparably infrequent. Delving deeper into this discrepancy could provide valuable insights and lessons from this outbreak and how Aneityum managed to prevent and control them.
- Finally, exploring how many samples could have originated from relapses might be worthwhile. This direction could present an intriguing line of inquiry for subsequent research if it is technically feasible. For clarity, I'm saying this as a possibility of future research, not to be addressed in this work.

A minor comment:

- In line 208, the authors wrote about 'intervening period'. I'm not sure which specific period was referred to by this.

[1] Kaneko A, Taleo G, Kalkoa M, Yamar S, Kobayakawa T, Björkman A. Malaria eradication on islands. *The Lancet*. 2000 Nov 4;356(9241):1560-4.

Reviewer #3 (Remarks to the Author):

This study typed samples from a *Plasmodium vivax* outbreak on Aneityum Island, Vanuatu, after malaria had been eliminated.

Of 219 samples positive, 69 were successfully typed for 7 microsatellite markers (out of an initial panel of 14 markers). The genotyping methods used are outdated. The *P. vivax* population in the south Pacific is well characterized using microsatellites. More advanced methods such as genome-wide SNP panels, multiplexed amplicon sequencing, or selective whole-genome sequencing have been published. Such methods would also allow for more advanced analytical approaches in addition to those included in the manuscript (which includes *Fst* values, PCA, STRUCTURE analysis, etc.). Given the same authors types the same samples before using amplicon sequencing of two markers, it is unclear what can be gained with a panel of 7 microsatellites .

In line with previous studies which typed the same samples using amplicon sequencing of two other markers, and in line with travel histories of infected individuals, the study found reduced genetic diversity and high similarity to parasites on neighboring Tanna island. No new insights into the origin of the outbreak are presented.

Minor comments:

Line 45: Is there a typo/missing word? I don't understand this sentence.

Line 100 hints that other species than *P. vivax* were found. This is crucial, if indeed multiple species were present, then certainly it can't be introduction of a single parasite.

Dear Dr. Walker and reviewers,

We thank you for your careful and thorough review of our manuscript and insightful comments on important points that needed to be included or addressed in our original submission. We greatly appreciate the opportunity to revise and resubmit our manuscript. Below are our point-by-point responses to the reviewers' comments.

Response to the Reviewer 1

>> PCR amplification of at least two nuclear loci used for DNA quality verification failed in 72 of the 182 mitochondrial *cox3* PCR-positive *P. vivax* samples: does this raise any concern about the performance (false positives?) of the *cox3* PCR?

Gene copy number is a main determinant of molecular diagnostic performance. We target the mitochondrial *cox3* gene to detect *P. vivax* infections because of the high gene copy number (20 to 150 copies in each parasite) that greatly enhances diagnostic sensitivity, especially for samples with low parasitemia. In contrast, the nuclear microsatellite markers used in this study are single-copy loci. Since we intended to use multiple microsatellite markers to characterize parasite genetic diversity, we set the threshold of successful amplification of at least two single-copy nuclear loci in the DNA quality check step. Of the 72 samples that did not meet such a threshold, 23 successfully amplified one nuclear locus. Of the 182 *cox3* positive samples, 133 (73.1%) were also positive by PCR of one or more nuclear loci. While we cannot definitively prove that there were no false positives with our *cox3* PCR, the proportion (26.9% or 49/182) of *cox3* positive but nuclear marker negative samples in this study is in line with the 30% PCR efficiency we reported previously in a comparison of the *cox3* and the standard 18S PCR³¹.

We added the following in the discussion to explain the difference in PCR efficiencies due to the difference in gene copy numbers between *cox3* and microsatellite markers (**lines 218-227**).

“Of the 182 *P. vivax*-positive samples by the mitochondrial *cox3* PCR, 49 (26.9%) failed to amplify any nuclear loci while 23 (12.6%) amplified only one nuclear locus. These 72 samples were deemed to have low DNA concentrations and were excluded from microsatellite marker analysis (Fig. 2). Molecular diagnostic yields are mainly determined by blood volume analysed and the gene copy number of the amplified molecular marker³⁰. The discrepancy in PCR amplification efficiencies in our study likely reflected such a difference in gene copy numbers between the mitochondrial *cox3* and the nuclear microsatellite markers. Within each parasite, there are 20 to 150 copies of the mitochondrial *cox3* gene³¹, but only one copy of nuclear

microsatellite markers, making the former an ideal target for PCR detection of *Plasmodium* infections.”

>> Adding a dispersion measure of the mean number of allele differences between paired samples would make more robust the claims of lower mean in the 2002 Tanna and 2002 Aneityum populations than in the 1999-2000 Aneityum and the 2000 non-Aneityum populations.

We added standard deviation as a dispersion measure of the mean number of allele differences between paired samples in table 3 and the result section (**lines 133-134**).

“The mean was lower in the 2002 Tanna (mean \pm SD: 2.857 ± 1.267) and 2002 Aneityum (2.091 ± 1.562) populations than in the 1999-2000 Aneityum (4.238 ± 0.865) and the 2000 non-Aneityum (4.562 ± 0.538) populations.”

>> Check line 45 (some letters are missing)

Some words were accidentally deleted during the formatting of the manuscript. The sentences have been revised as follows (**p.2, lines 46-48**),

“Islands provide natural ecological experiments with considerable potential for malaria intervention studies⁸ and some have demonstrated early success towards malaria elimination^{9,10}. Vanuatu is an archipelago of 82 islands spanning 1,176 km in the South Pacific.”

Response to Reviewer 2

>> In Figure 3, I noticed an intriguing observation where a sample from the 2002 Aneityum samples closely aligns with the 1999-2000 Aneityum and 2000 non-Aneityum clusters. This particular observation doesn't seem to be emphasised in the manuscript. Do the authors consider the proximity of this sample to be insignificant? If so, elaborating on the reasons might be important, given this observation might explain the connections between the 2002 and 2000 populations.

We were less confident about the relationship among these three lineages than the relationship between haplotypes 26 and 42 shared between Aneityum and Tanna (they share the same alleles in four of five loci genotyped). Nonetheless the observation that these three lineages showed genetic affinity in both neighbor-joining tree and PCA warrants further examination. We added the following in the discussion (**lines 173-180**),

“Phylogenetic analysis (Fig. 3A) and PCA (Fig. 3C) also indicated genetic affinity among one 2002 Aneityum sample and one 1999/2000 Aneityum and one 2000 non-Aneityum samples. The 1999/2000 Aneityum sample was derived from a male child who had moved to Aneityum from Pentecost. The child was diagnosed with *P. vivax* in May 1999 and again in July 1999 during our survey. The 2000 non-Aneityum sample was derived from a male adult church meeting attendee from Tanna. Not only did these infections exemplify the vulnerability of Aneityum to *P. vivax* importation, but they also suggested the possibility that one of the seeds for the 2002 outbreak may have been introduced in 1999 or 2000.”

>>While the authors have underscored the significance of relapses in *P. vivax* control in the introduction, there is limited discussion on the potential role of relapses in seeding the 2002 outbreak. The 2000 study [1] indicated that relapses could be observed up to 5 years after the initial infection. Could these long-latency relapses from prior infections have been a contributing factor to the onset of the outbreak?

Yes, long-latency relapses from infections acquired years ago on Aneityum or other islands in Vanuatu could have contributed to the outbreak in 2002. We added such possibility in the discussion (**lines 162-165**),

“Notably we previously indicated the potential of *P. vivax* relapses occurring up to five years after the initial infection¹⁶, raising the possibility that the 2002 outbreak was partially seeded from

long-latency relapses from infections initially acquired years before.”

>> Taking into account the immunity resulting from previous infections (predominantly in adults), the frequency of travel (primarily by adults, I assume), which together might result in importation of asymptomatic cases, and coupled with the potential long-latency of relapse: How probable is it, in the authors' opinion, that the outbreak's seeding occurred in the past, possibly linked to travels beyond Tanna (explaining the minimal connection with Tanna samples) and was potentially triggered by a recent relapse?

This is an expanded and more nuanced discussion of the point raised in the previous comment. We acknowledge that the scenario presented by the reviewer is possible, though we have no way of quantifying that probability given the data we have. We added in the discussion the following, **(lines 245-248)**,

“Since *P. vivax* infections in adults were sub-microscopic due to persisting immunity acquired before malaria elimination and undetectable by diagnostics available locally, coupled with the potential of long-latency relapse, the 2002 outbreak's seeding event could have occurred years ago and the outbreak itself could have been triggered by a recent relapse.”

>> Finally, exploring how many samples could have originated from relapses might be worthwhile. This direction could present an intriguing line of inquiry for subsequent research if it is technically feasible. For clarity, I'm saying this as a possibility of future research, not to be addressed in this work.

We agree with the reviewer that the ability to distinguish between primary infections and relapses in *P. vivax* cases will be tremendously useful for control and elimination. We added in the discussion the following **(lines 249-252)**,

“Determining the proportion of infections resulting from relapse during *P. vivax* outbreaks can inform the most optimal response strategy, however distinguishing between primary infections and relapses in settings such as Aneityum or Vanuatu, where conditions conducive to transmission remain, is difficult.”

>> What is the level of receptivity of Aneityum to malaria transmission? An outbreak resulting in 10% of the population being infected at its tail end suggests that the island remains highly receptive to transmission. Were there subsequent outbreaks in the following years? It's curious

that while the introduction of cases seemed to occur somewhat frequently, and the island appeared highly receptive to transmission, actual outbreaks were comparably infrequent. Delving deeper into this discrepancy could provide valuable insights and lessons from this outbreak and how Aneityum managed to prevent and control them.

Receptivity is a difficult measure to quantify. We preface our response by stating that we have no data on vector density, sporozoite rate, and biting rate during the outbreak in 2002. We believe that the level of receptivity to malaria transmission was fairly high during our survey in July 2002. Our genetic data showed that most parasites were genetically similar, which is consistent with the rapid expansion of a limited number of parasite clones via mosquito transmission. We detailed the subsequent community responses to the outbreak, and how high ITN usage among community members might have curtailed the extent of the outbreak (lines 253-263).

“Although we lack entomological data, the observation that most *P. vivax* isolates during the 2002 outbreak were genetically very similar suggests that transmission by *Anopheles* mosquitoes played a major role in sustaining the outbreak. Therefore, it can be inferred that the level of receptivity was probably very high in 2002. In response, a second round of MDA was implemented, targeting individuals under 20 years of age. At the same time, provisions of ITNs were strengthened, coupled with a robust malaria health promotion programme to raise awareness of the continuous risk of malaria resurgence among local communities¹⁵. Subsequent annual cross-sectional surveys revealed low prevalence (1.9% to 3.9%) of *P. vivax* infections by PCR between 2003 and 2007. Since 2010, no *Plasmodium* infections have been detected by microscopy and PCR³². Nonetheless, 74.1% and 67.4% of the island’s residents reported ITN use in 2016 and 2019, respectively (unpublished data). We believe that high ITN usage likely suppressed malaria receptivity on Aneityum Island.”

>> In line 208, the authors wrote about ‘intervening period’. I’m not sure which specific period was referred to by this.

We meant the period between the start of the outbreak in early 2002 and our survey in July 2002. For clarity, we revised the sentence to the following (lines 234-236),

“Additional but undetected parasite reintroduction events between the onset of the outbreak in early 2002 and our survey in July 2002 could have obscured or even replaced the original *P. vivax* clone that triggered the outbreak.”

Response to Reviewer 3

>> More advanced methods such as genome-wide SNP panels, multiplexed amplicon sequencing, or selective whole-genome sequencing have been published. Such methods would also allow for more advanced analytical approaches in addition to those included in the manuscript (which includes *Fst* values, PCA, STRUCTURE analysis, etc.). Given the same authors types the same samples before using amplicon sequencing of two markers, it is unclear what can be gained with a panel of 7 microsatellites.

We fully acknowledge the reviewer's concern. More advanced methods and analytical approaches using high-resolution data may be helpful. However, most of our samples were from asymptomatic and submicroscopic infections mixed with human DNA because they were extracted from DBS. Our previous experience with DBS samples from submicroscopic *P. falciparum* infections in Kenya^{#1} suggests that our samples from Vanuatu are unlikely to be amenable to existing advanced genomic methods to characterize parasite genetic diversity. Given these constraints, we chose to genotype some of the most polymorphic markers using technologies and methods available to us.

Although we fully understand the importance of further advancing malaria genomics from an academic standpoint, there is some discussion that advanced malaria genomics may have several gaps for “real-world malaria control and elimination strategy”^{#2} and may not be necessary for malaria control or even its elimination^{#3}.

We also want to point out that the samples analysed in this manuscript differ those used in the previous paper. In our previous work, we obtained DNA sequences from only microscopically positive samples, whereas in the current study the majority of data comes from submicroscopic (PCR positive but microscopy negative) infections. Previously we speculated that infected church meeting attendees in 2000 could represent the sources of imported parasites that triggered the outbreak two years later. Those samples were not analysed in the previous paper but are included here.

#1 Osborne A, Manko E, Takeda M, Kaneko A, Kagaya W, Chan C, Ngara M, Kongere J, Kita K, Campino S, Kaneko O, Gitaka J, Clark TG. Characterizing the genomic variation and population dynamics of *Plasmodium falciparum* malaria parasites in and around Lake Victoria, Kenya. *Sci Rep* **11**, 19809 (2021). Doi: 10.1038/s41598-021-99192-1

#2 Neafsey, D.E., Taylor, A.R. & MacInnis, B.L. Advances and opportunities in malaria population genomics. *Nat Rev Genet* **22**, 502–517 (2021). <https://doi.org/10.1038/s41576-021-00349-5>

#3 Neafsey DE, Volkman SK. Malaria Genomics in the Era of Eradication. Cold Spring Harb Perspect Med. 2017 Aug 1;7(8):a025544. doi: 10.1101/cshperspect.a025544. PMID: 28389516; PMCID: PMC5538406.

Based on the above, the following text was added (**lines 227-231**).

“Low *P. vivax* nuclear DNA concentrations in most samples precluded using advanced methods and analytical approaches to gain a more comprehensive understanding of the population genomic diversity of *P. vivax* parasites. Nevertheless, there is some discussion that advanced malaria genomics may have several gaps for an immediate “real-world malaria control and elimination strategy”³² and may not be necessary for malaria control or even its elimination³³.”

>> In line with previous studies which typed the same samples using amplicon sequencing of two other markers, and in line with travel histories of infected individuals, the study found reduced genetic diversity and high similarity to parasites on neighboring Tanna Island. No new insights into the origin of the outbreak are presented.

We believe our findings are valuable, at the least, in building a baseline that would be useful for formulating malaria control and elimination policies in Vanuatu and beyond.

>> Line 45: Is there a typo/missing word? I don't understand this sentence.

See response to comment by reviewer 1.

*>> Line 100 hints that other species than *P. vivax* were found. This is crucial, if indeed multiple species were present, then certainly in can't be introduction of a single parasite.*

The statement in line 100 refers to the detection of *Plasmodium* spp. in all samples collected from different islands in Vanuatu from 1999 to 2002. On Aneityum Island, only *P. vivax* was detected during the entire study period. We added the following for clarity, (**lines 102-105**)

“Throughout the study period, neither microscopy nor PCR detected any species other than *P. vivax* on Aneityum Island, and the prevalence of *P. vivax* infection by PCR increased from 0.8% (6/709) in 1999 to 10.1% (77/759) in 2002 (Table 1).”

***Note to editor**

In the original submission, the colours representing samples from 1999/2000 Aneityum and 2000 non-Aneityum in Fig. 3C were reversed and have been corrected in this revised submission. Minor edits in the text are highlighted in green.

REVIEWERS' COMMENTS:

Reviewer #1 (Remarks to the Author):

My comments and questions were correctly addressed. I do not have any further comment.

Re-review in response Reviewer 3's comments to authors:

I would be considered myself satisfied with the responses given to comments by reviewer 3, which were mainly about:

- The use of more advanced genetic methods: authors argue are not feasible given low parasite densities in samples)
- no new insights into the origin of the outbreak are presented: authors argue that findings are valuable in building a baseline for formulating malaria control and elimination policies in Vanuatu and beyond.

Reviewer #2 (Remarks to the Author):

I would like to thank the authors for their efforts in addressing my comments. I am happy with authors' comments and their additions/revisions to the manuscript in regards to my initial review, and, hence, have no additional comments to add.